# Smart(Sampling)Augment: Optimal and Efficient Data Augmentation for Semantic Segmentation

**Misgana Negassi** [1,2,*,†] , **Diane Wagner** [1,†] **and Alexander Reiterer** [1,2]

1   Institute for Sustainable Systems Engineering INATECH, Albert Ludwigs University of Freiburg,
    79110 Freiburg, Germany; wagnerd@cs.uni-freiburg.de (D.W.); alexander.reiterer@ipm.fraunhofer.de (A.R.)
2   Fraunhofer Institute for Physical Measurement Techniques IPM, 79110 Freiburg, Germany
*   Correspondence: misgana.negassi@ipm.fraunhofer.de
†   These authors contributed equally to this work.

**Abstract:** Data augmentation methods enrich datasets with augmented data to improve the performance of neural networks. Recently, automated data augmentation methods have emerged, which automatically design augmentation strategies. The existing work focuses on image classification and object detection, whereas we provide the first study on semantic image segmentation and introduce two new approaches: *SmartAugment* and *SmartSamplingAugment*. SmartAugment uses Bayesian Optimization to search a rich space of augmentation strategies and achieves new state-of-the-art performance in all semantic segmentation tasks we consider. SmartSamplingAugment, a simple parameter-free approach with a fixed augmentation strategy, competes in performance with the existing resource-intensive approaches and outperforms cheap state-of-the-art data augmentation methods. Furthermore, we analyze the impact, interaction, and importance of data augmentation hyperparameters and perform ablation studies, which confirm our design choices behind SmartAugment and SmartSamplingAugment. Lastly, we will provide our source code for reproducibility and to facilitate further research.

**Keywords:** data augmentation; hyperparameter optimization; semantic segmentation

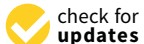



## 1. Introduction

In many real-world applications, only a limited amount of annotated data is available, which is particularly pronounced in medical imaging applications, where expert knowledge is indispensable to annotate data accurately [1,2]. Given insufficient training data, deep learning methods frequently overfit and fail to learn a discriminative function that generalizes well to unseen examples [3]. *Data augmentation* is an established approach that improves the generalization of neural networks by adjusting the limited available data to achieve more and diverse samples for the network to train on. In most cases, additional data are constructed by simply applying label-preserving transformations to the original data. In image processing, for instance, these can be simple geometric transformations (e.g., rotation), color transformations (e.g., contrast adjustments), or more complex approaches such as CutMix [4], Cutout [5], and Mixup [6]. Data augmentation has been applied to various areas, such as image classification [6], object detection [7], and semi-supervised learning [8] and segmentation [9]. This work provides a first and extensive study on automated data augmentation for semantic segmentation on different and diverse datasets.

Data augmentations used in practice are mostly simple and easy to implement. Despite this simplicity, the choice of augmentations is crucial and requires domain knowledge. Recently, automated data augmentation methods were proposed that learn optimal augmentation policies from data without the need for domain knowledge. These approaches improve performance over manually designed data augmentation strategies commonly used across different domains and datasets [10–13].

The main focus of existing research in automated data augmentation is image classification [10,12], with a particular blind spot being dense prediction tasks such as semantic segmentation. Furthermore, these methods either use complicated proxy tasks to learn an optimal augmentation strategy [10] or optimize the augmentation operations without taking the type of augmentation applied and the probability of their application into account [12].

In this work, we introduce two novel data augmentation methods, *SmartAugment* and *SmartSamplingAugment* with key focus on diverse semantic segmentation applications: medical imaging (RaVeNNa, EM), bridge inspection (ErFASst), and autonomous driving (KITTI). SmartAugment uses Bayesian Optimization [14,15] to optimize data augmentation strategies and outperforms the previous state-of-the-art methods (see Table 1) across all semantic segmentation tasks we consider. In contrast to existing approaches, we define a separate set of each color and geometric data augmentation operations, search for their optimal number of operations and magnitudes, and further optimize a probability $P$ of applying these augmentations.

**Table 1.** Test mean Intersection over Union (IoU) in percentage for different algorithms on semantic segmentation datasets. SmartAugment outperforms all other data augmentation strategies across all datasets. SmartSamplingAugment competes with the previous state-of-the-art approaches and outperforms TrivialAugment, a comparably cheap method. For DefaultAugment, TrivialAugment, and SmartSamplingAugment, we evaluated each experiment three times using different seeds to obtain the mean performance. For RandAugment++ (an extended version of RandAugment) and SmartAugment, we took the mean test IoU over the three best performing validation configurations. Please note that DefaultAugment represents the baseline, and the higher the value, the better the performance. # iterations refers to the number of BO iterations completed to find best configuration.

| Dataset | Default | Rand++ | Trivial | Smart | SmartSampling |
|---|---|---|---|---|---|
| KITTI | 65.07 | 67.19 | 64.82 | **68.84** | 66.53 |
| RaVeNNa | 88.37 | 90.71 | 90.53 | **91.00** | 90.72 |
| EM | 77.25 | 78.83 | 78.15 | **79.04** | 78.42 |
| ErfASst | 67.01 | 68.75 | 66.79 | **73.72** | 70.24 |
| # Iterations | 1 | 50 | 1 | 50 | 1 |

SmartAugment performs well compared to existing approaches in performance and computational budget. However, it still requires multiple iterations to find the best augmentation strategy, which can be expensive for researchers with computational constraints. With this in mind, we develop a fast and efficient data augmentation method, SmartSamplingAugment, that has a competitive performance to current best methods and outperforms TrivialAugment [16], a previous state-of-the-art simple augmentation method. SmartSamplingAugment is a parameter-free approach that samples augmentation operations according to their weights, and the probability of application is annealed during training. We summarize our contributions in the following points:

- We provide a first and extensive study of data augmentation on different and diverse datasets for semantic segmentation.
- We introduce a new state-of-the-art automated data augmentation algorithm for semantic segmentation that outperforms previous methods with half of the computational budget. It optimizes the number of applied geometric and color augmentations and their magnitude separately. Furthermore, it optimizes the probability of augmentation, which is crucial according to our hyperparameter importance analysis.
- We present a novel parameter-free data augmentation approach that weighs the applied data augmentation operations and anneals their probability of application. Our method is competitive with the previous automated data augmentation approaches and outperforms TrivialAugment, a cheap-to-evaluate method.

We will provide our source code: https://github.com/mvg-inatech/SmartAugment. (accessed on 1 April 2022).

## 2. Related Work

Data augmentation has been shown to have a considerable impact, particularly on computer vision tasks. Simple augmentation methods such as random cropping, horizontal flipping, random scaling, rotation, and translation have been effective and popular for image classification datasets [17–20]. Other approaches add noise or erase part of an image [5,21] or apply a convex combination of pairs of images and their labels [6]. Other approaches use generative adversarial networks to generate new training data [22,23].

Automated augmentation methods focus on learning an optimal data augmentation strategy from data [10,12]. Many recent methods define a set of data augmentations and their magnitude, where the best augmentation strategy is automatically selected. AutoAugment [10] uses a search algorithm based on reinforcement learning to find the best data augmentation policy with a validation accuracy as the reward. The search space consists of policies which in turn, have many sub-policies. Each sub-policy contains two augmentation operations, their magnitude, and a probability of application. A sub-policy is selected uniformly at random and applied to an image from a mini-batch. This process has high computational demands; therefore, it is applied on a proxy task with a smaller dataset and model. The best-found augmentation policy is then applied to the target task.

Population-Based Augmentation (PBA) [24] uses a population-based training algorithm [13] to learn a schedule of augmentation policies at every epoch during training. The policies are parameterized to consist of the magnitude and probability values for each augmentation operation. PBA randomly initializes and trains a model with these different policies in parallel. The weights of the better-performing models are cloned and perturbed with noise to make an exploration and exploitation trade-off. The schedule is learned with a child model and applied to a larger model on the same dataset.

Fast AutoAugment [25] speeds up the search for the best augmentation strategy with density matching. This method directly learns augmentation policies on inference time and tries to maximize the match of the distribution between augmented and non-augmented data without the need for child models. The idea is that if a network trained on real data generalizes well on augmented validation data, then the policy that produces these augmented data will be optimal. In other words, the policy preserves the label of the images, thus the distribution of the real data.

Adversarial AutoAugment [26] optimizes a target network and augmentation policy network jointly on target task in an adversarial fashion. The augmentation policy network generates data augmentations policies that produce hard examples, therefore increasing the target network's training loss. The hard examples force the target network to learn more robust features that improve its generalization and overall performance.

RandAugment [12] uses a much reduced search space than AutoAugment and optimizes two hyperparameters: the number of applied augmentations and the magnitude. RandAugment tunes these parameters with a simple Grid Search [27] on the target task, therefore, removes the need for a proxy task as is the case in AutoAugment [10]. The authors argue that this simplification helped the strong performance and efficiency of their approach.

TrivialAugment [16] samples one augmentation from a given set of augmentations and its magnitude uniformly at random and applies on a given image. This method is efficient, parameter-free, and competes with RandAugment [12] in performance for image classification.

In this work, we introduce two novel (automated) data augmentation methods for semantic segmentation: SmartAugment and SmartSamplingAugment. With hyperparameter optimization, SmartAugment finds optimal data augmentation strategy and SmartSamplingAugment's efficient and parameter-free approach competes with the previous state-of-the-art methods.

## 3. Methods

In this section, we present our data augmentation algorithms: SmartAugment and SmartSamplingAugment. Similar to previous methods, namely RandAugment and TrivialAugment, we define a set of color and geometric augmentations along with their magnitudes as shown in Table 2. We describe our algorithms in detail in the following subsections.

**Table 2.** Detailed overview of data augmentation operations and their magnitude ranges. We use the same augmentations as in RandAugment paper [12]. * The Identity operation only belongs to this list for the RandAugment and TrivialAugment approaches.

| Color Ops | Range | Geometric Ops | Range |
|---|---|---|---|
| Sharpness | (0.1, 1.9) | Rotate | (0, 30) |
| AutoContrast | (0, 1) | ShearX | (0.0, 0.3) |
| Equalize | (0, 1) | ShearY | (0.0, 0.3) |
| Solarize | (0, 256) | TranslateX | (0.0, 0.33) |
| Color | (0.1, 1.9) | TranslateY | (0.0, 0.33) |
| Contrast | (0.1, 1.9) | Identity * | |
| Brightness | (0.1, 1.9) | | |

### 3.1. Smartaugment

SmartAugment optimizes the number of sampled color and geometric augmentations and their magnitude separately (see Figure 1b and Algorithm 1). Having these distinct sets of augmentations allows control over the type of applied augmentation instead of optimizing the total number of sampled augmentations and their magnitude collectively. SmartAugment also optimizes a parameter that determines the probability of applying data augmentations $P$ instead of having the Identity operation in the augmentation list, as done by recent approaches.

SmartAugment uses Bayesian Optimization (BO) [15] to search for the best augmentation strategy. The space of augmentation strategies include the following parameters: number of color augmentations $N_C$, number of geometric augmentations $N_G$, color magnitude $M_C$, geometric magnitude $M_G$, and probability of applying augmentations $P$. These hyperparameters are optimized with the BO algorithm until a given budget is exhausted. Once BO chooses the augmentation parameters, the augmentations are sampled randomly without replacement for each epoch and image from the given list of augmentation operations as listed in Table 2. RandAugment, in contrast, samples with replacement and therefore allows sampling the same augmentation several times for the same image.

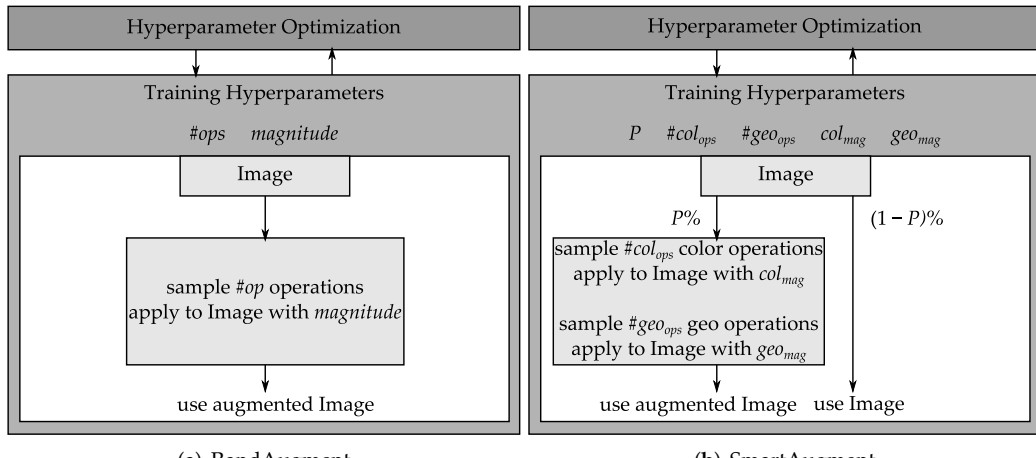

**(a)** RandAugment          **(b)** SmartAugment

**Figure 1.** Comparison of RandAugment and SmartAugment.

---

**Algorithm 1:** Pseudocode for SmartAugment.

**Input:** Data $D$,
List of color augmentations $C_{LIST}$,
List of geometric augmentations $G_{LIST}$

1 **for** *each configuration* **do**
2    Select 5 hyperparameters via BO:
3      1) # Color augmentations $N_C$ to sample,
4      2) # Geometric augmentations $N_G$ to sample,
5      3) Color magnitude $M_C$,
6      4) Geometric magnitude $M_G$,
7      5) Probability $P$ of applying augmentations
8    **for** *each epoch* **do**
9      **for** *each image $I$ in $D$* **do**
10        Sample *var* uniformly from $[0, 1]$
11        **if** *var > P* **then**
12          use $I$ ;                  `// do not augment`
13        **else**
14          $C$ := random sample $N_C$ ops from $C_{LIST}$
15          $G$ := random sample $N_G$ ops from $G_{LIST}$
16          $I_{AUG}$ := apply $C$ with $M_C$
17              and $G$ with $M_G$ to $I$
18          use $I_{AUG}$

---

### 3.2. SmartSamplingAugment

The number of sampled augmentations in SmartSamplingAugment, a tuning-free and computationally efficient algorithm, is fixed to two augmentation operations, and the magnitude is sampled randomly from the interval [5, 30] (see Figure 2b and Algorithm 2). These design choices are based on our preliminary experiments and seem to generalize well to unseen datasets. SmartSamplingAugment samples augmentations with a probability derived from the weights, which we set based on an ablation study for image classification on CIFAR10 from RandAugment [12]. In this study [12], the average improvement in performance is computed when a particular augmentation operation is added to a random subset of augmentations. We selected the augmentations with a positive average improvement and transformed this value into probabilities, by which we define the weights.

In SmartSamplingAugment, we linearly anneal the parameter $P$, that determines the probability of applying data augmentations, from 0 to 1, increasing the percentage of applying augmentation over the whole training epochs. That way, the model first sees the original data in the early epochs and encounters more variations as the training progresses.

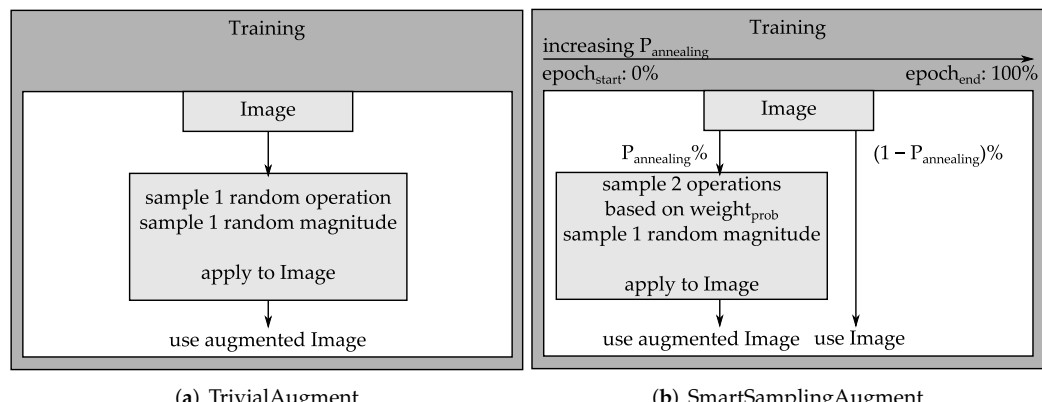

(**a**) TrivialAugment           (**b**) SmartSamplingAugment

**Figure 2.** Comparison of TrivialAugment and SmartSamplingAugment.

---

**Algorithm 2:** Pseudocode for SmartSamplingAugment.

---

**Input:** Data $D$,
List of augmentations $A := [a_1, a_2, \ldots, a_{-1}]$,
Weights $W := [w_{a_1}, w_{a_2}, \ldots, w_{a_{-1}}]$

1 **for** *each epoch* **do**
2     Update $P$ ;                                     `// P is linearly annealed`
3     **for** *each image I in D* **do**
4         Sample *var* from [0, 1]
5         **if** *var > P* **then**
6             use $I$ ;                              `// do not augment`
7         **else**
8             $A_W$ := random sample 2 ops $a_i, a_j \in A$ with weights $w_i, w_j \in W$
9             $M$ := random sample magnitude from [5,28]
10             $I_{AUG}$ := apply $A_W$ with $M$ to $I$
11             use $I_{AUG}$

---

## 4. Experiments and Results

In this section, we empirically evaluate and analyze the performance of SmartAugment and SmartSamplingAugment on several datasets and compare it to the previous state-of-the-art approaches. Furthermore, we investigate the impact, interaction, and importance of the optimized data augmentation hyperparameters.

### *4.1. Experimental Setup*

#### 4.1.1. Default and TrivialAugment

For completeness, we include in our experiments a "standard" augmentation strategy, a strategy based on augmentations often manually selected by researchers , we dubbed *DefaultAugment* and use it as our baseline. This default augmentation strategy is inspired by semantic segmentation literature [28,29] and uses the following standard augmentations: horizontal flipping ($p_{flip} = 0.5$), random rotation ($range = [-45, 45]$), random scaling ($range = [-0.35, 0.35]$), where $p_{flip}$ represents the probability of applying this particular augmentation. Furthermore, we extended another recent method, TrivialAugment (see Figure 2a), for semantic segmentation and integrated it in our experiments.

#### 4.1.2. RandAugment++

Classical RandAugment [12] uses simple Grid Search [27] to optimize its hyperparameters. Evaluating the full grid of classical RandAugment would lead to evaluate nearly 100 iterations ($31 \times 3$ iterations: magnitude in the range of [0, 30] and the number of operations in the range of [1, 3]) which is computationally very expensive. Therefore, we decided to implement an updated version of RandAugment, which we call *RandAugment++* (see Figure 1a) that uses the same algorithm but optimizes its hyperparameters with Random Search. Random Search is known to perform better than Grid Search [30] and its number of iterations is not limited to the size of the grid in the search space. Furthermore, using Random Search enabled us to reduce the number of iterations and increase the search space for RandAugment++ with less computational costs. We analyzed the performance of RandAugment++ with different operations on the EM dataset and found out that constraining the number of applied operations to three is not optimal (see Figure 3). From this observation, we increase the upper limit for the number of applied operations from 3 to 16, which denotes the total number of augmentations in the list we sample from. To ensure comparability between RandAugment++ and SmartAugment, we use the same computational budget of 50 iterations for both methods. We show in an ablation study (see Section 4.4) that RandAugment++ is a better choice than the classical RandAugment.

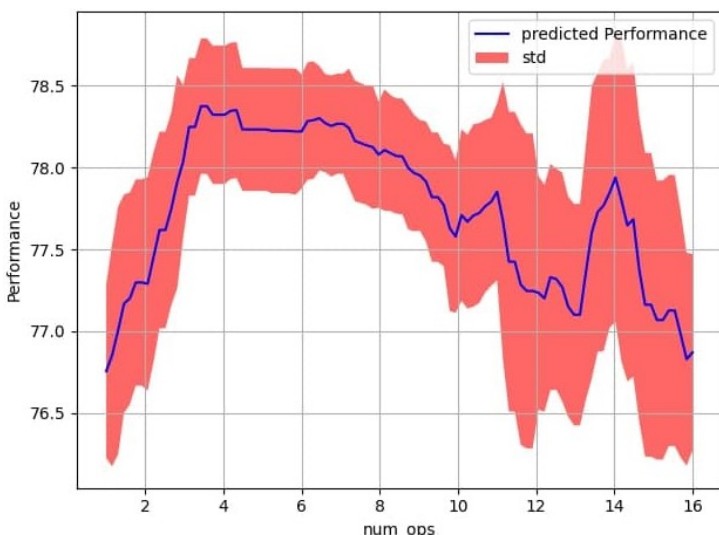

**Figure 3.** Performance (mean IoU) analysis for different numbers of operations for RandAugment++ on EM. These results indicate that the number of applied operations should optimally not be limited to three, as in classical RandAugment.

### 4.1.3. Training Setup

For all experiments, we use the U-Net architecture [31] to train the models and split the datasets into training, validation, and test set. To find good fitting training hyperparameters (e.g., learning rate and weight decay) for our in-house datasets, we performed Random Search over ten configurations until model convergence. As a preprocessing step, we apply with 50% probability either random crop or downsize operations before passing the data to the different augmentation strategies for efficient memory and computing use. For KITTI and EM datasets, we use a similar training setup as in [28]. To speed up memory intensive processes, we use mixed precision training with 16 bits. For our experiments, we made use of four GeForce GTX 1080 GPUs. For better reproducibility, we list the training parameters for each of the datasets for a detailed view in Table 3.

**Table 3.** Training parameters for each dataset. Train, val, test denote the data split used during training.

| Dataset | Resolution | Batch Size | Learning Rate | Epochs | # Data | #Train | #Val | #Test |
|---------|-----------|-----------|---------------|--------|--------|--------|------|-------|
| KITTI | $185 \times 612$ | 4 | 0.001 | 4000 | 200 | 140 | 30 | 30 |
| RaVeNNa | $180 \times 180$ | 3 | 0.001 | 2000 | 1684 | 1107 | 216 | 361 |
| EM | $512 \times 512$ | 2 | 0.01 | 500 | 30 | 20 | 5 | 5 |
| ErfASst | $864 \times 864$ | 2 | 0.05 | 5000 | 50 | 30 | 10 | 10 |

Furthermore, we performed early stopping on the validation set. To save computing and still obtain enough samples on the validation set for early stopping, we evaluate every 10% of the total epochs on the validation set. This ensures that for each dataset, independent of the number of epochs needed until convergence, the number of epochs evaluated on the validation set is proportional to the total number of epochs. We run these experiments three times for the different data augmentation approaches and take the mean of the test IoU to ensure a fair comparison. In the case of RandAugment and SmartAugment, we evaluated 50 configurations for each method and report the mean test IoU of the three best performing configurations on the validation set.

For all our experiments, we use Stochastic Gradient Descent (SGD) optimizer [32] and Cosine Annealing [33] as our learning rate scheduler, and anneal the learning rate over the total number of epochs.

#### 4.1.4. Datasets

We evaluate all approaches on four datasets with pixel-level annotated images from diverse semantic segmentation applications. ErFASst is a bridge inspection dataset with 50 images and two classes used for crack detection (Figure 4a). We use KITTI [34], a popular autonomous driving dataset consisting of 200 images with 19 classes (Figure 4b). RaVeNNa [1], is a cystoscopic medical imaging dataset comprises 1684 images with seven classes that is used in detecting artifacts such as tumors in human bladder (Figure 5a). EM [35] is a brain electron microscopy dataset of 30 images derived from a 2D segmentation challenge dataset and consists of two classes (Figure 5b). To achieve meaningful results, these datasets differ in size, resolutions, and type of images (RGB natural images, Grayscale, see Table 3). Since RaVeNNa and ErFASst are highly class-imbalanced datasets, we use a weighted cross-entropy loss during training. The weights are computed beforehand with inverse frequency of the number of pixels belonging to a specific class in the training set.

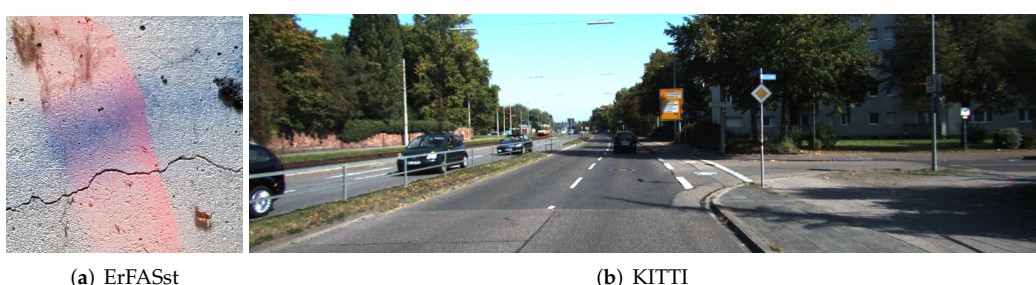

(a) ErFASst                                                        (b) KITTI

**Figure 4.** Infrastructure mapping datasets.

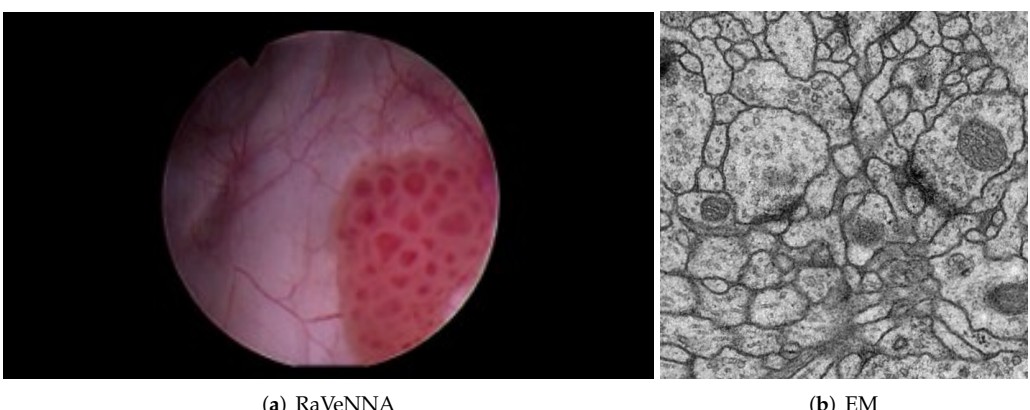

(a) RaVeNNA                                                        (b) EM

**Figure 5.** Biomedical datasets.

#### 4.2. Comparison to the State-of-the-Art

In Table 1, we compare our methods, SmartAugment and SmartSamplingAugment, to the aforementioned data augmentation methods as well as to our baseline, Default-Augment, a basic augmentation strategy that is commonly used in semantic segmentation literature [28,29]. SmartAugment outperforms the previous state-of-the-art methods across all datasets, while SmartSamplingAugment competes with the previous state-of-the-art methods and outperforms the comparably cheap augmentation method, TrivialAugment. Moreover, SmartSamplingAugment outperforms RandAugment++ on half of the datasets, even though the latter has 50 times more budget.

#### Approximate Analysis of Compute Costs

We calculate the computing costs for each dataset for all the experiments done with GeForce GTX 1080. In Table 4 we list our cost estimates that each method requires until convergence.

**Table 4.** Approximate estimate of computing costs (in hours) for each dataset and augmentation approach. The costs are the time required until the maximum number of epochs is reached. Due to computational resource constraints, we run Smart, Rand++, $Rand_{classic}$ with four GPUs in parallel. $Rand_{classic}$ is RandAugment that uses GridSearch for optimization.

| Dataset | Default | Rand++ | Trivial | Smart | Smart Sampling | $Rand_{classic}$ |
|---|---|---|---|---|---|---|
| KITTI | 12 h | 150 h | 12 h | 150 h | 12 h | 276 h |
| RaVeNNa | 13 h | 163 h | 13 h | 13 h | 163 h | 302 h |
| EM | 0.5 h | 6 h | 0.5 h | 6 h | 0.5 h | 11.6 h |
| ErfASst | 23 h | 287 h | 23 h | 287 h | 23 h | 537 h |
| # Iterations | 1 | 50 | 1 | 50 | 1 | 93 |

### 4.3. Analysis with fANOVA

We analyze the impact, interaction, and importance of augmentation hyperparameters across different datasets with fANOVA [36]. Moreover, we quantify and visualize the effect of different augmentation configurations on the overall model performance on the validation mean IoU metric.

#### 4.3.1. Impact of Hyperparameters across Different Datasets

The results in Figures 6 and 7 show that the optimal strategy of augmentation hyperparameters is dataset-specific and predominantly impacts the overall performance: As shown in Figure 6, applying many color operations with a high color magnitude to the data can be good for the EM dataset, but can have a detrimental effect on the performance of the KITTI dataset. There are areas in the augmentation space where it is sub-optimal to sample from for a particular dataset but are good for another one. Furthermore, Figure 7 indicates that the probability hyperparameter of applying data augmentations strongly varies across the datasets.

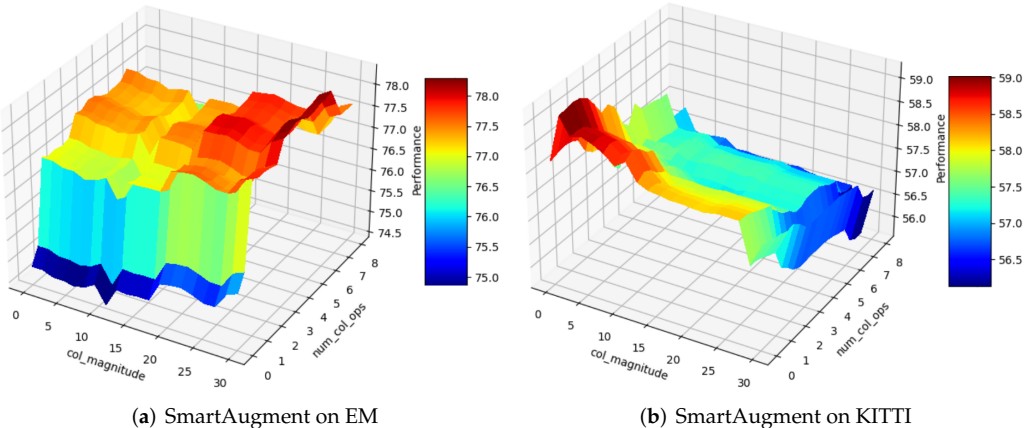

(**a**) SmartAugment on EM      (**b**) SmartAugment on KITTI

**Figure 6.** The impact of hyperparameters on different datasets based on performance metric mean IoU. This figure shows that the good values for each hyperparameter depend on the dataset. In this example, higher number of color ops and color magnitude is optimal for EM dataset but detrimental for KITTI dataset.

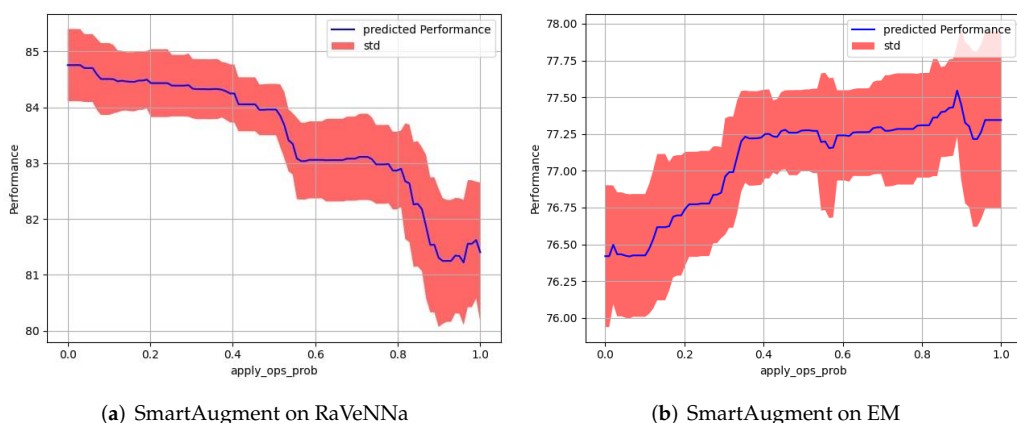

(**a**) SmartAugment on RaVeNNa　　　　　　　(**b**) SmartAugment on EM

**Figure 7.** Comparison of the probability hyperparameter of applying data augmentations. These results indicate that the EM dataset needs much more data augmentation than the RaVeNNa dataset. MeanIoU is used as performance metric.

### 4.3.2. Hyperparameter Interaction Analysis

Furthermore, we analyze the interaction of hyperparameters and their effect on the performance. As mentioned in Section 3, SmartAugment optimizes the color and geometric augmentations separately. The results in Table 1 and Figure 8 confirm our hypothesis that this is a good design choice. Looking more closely, the figure shows that for optimal performance, it does not suffice to optimize the total number of applied augmentations; rather, it is crucial to sample the right type of augmentation from the augmentation list carefully. For instance, according to Figure 8b, choosing four operations from the total number of augmentation seems to be the optimal choice for the KITTI dataset. However, according to Figure 8a, just sampling "blindly" four augmentation operations from the entire augmentation list might not always be a good choice. If we would pick four color augmentation operations and zero geometric augmentation operations, the performance would be significantly sub-optimal.

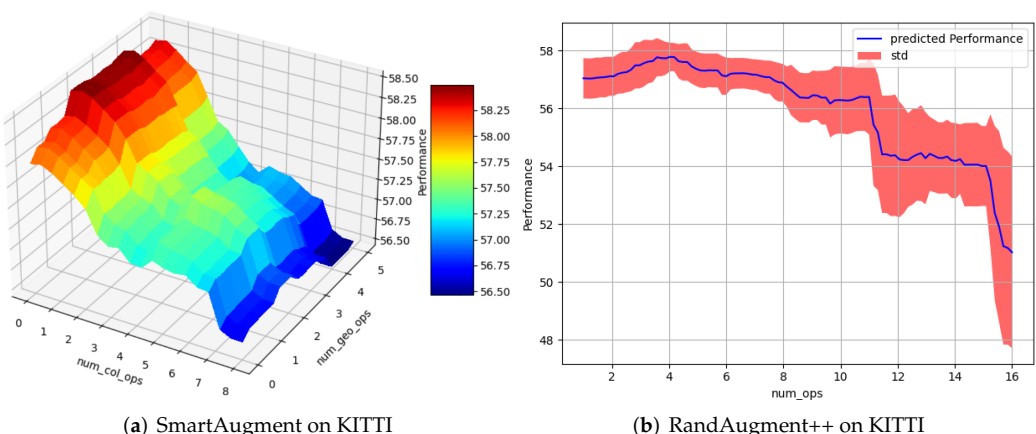

(**a**) SmartAugment on KITTI　　　　　　　(**b**) RandAugment++ on KITTI

**Figure 8.** Comparison considering the number of applied augmentations: RandAugment++ optimizes the total number of augmentations, whereas SmartAugment differs between the number of color augmentations and geometric augmentations. This figure shows that the performance of a total number of augmentations depends on the types of augmentations. Please note that mean IoU is used as a performance metric.

### 4.3.3. Hyperparameter Importance Study

In many algorithms that have a large hyperparameter space, only a few parameters are usually responsible for most of the performance improvement [36]. In this study, we use

fANOVA to quantify how much each hyperparameter contributes to the overall variance in performance. As we observe in Table 5, the importance of a specific hyperparameter strongly depends on the dataset. For instance, the geometric magnitude has a much higher impact on the KITTI dataset than other datasets. Moreover, the results from Table 5 show that optimizing the probability of application is an important design choice since this parameter is the most important one in half of the datasets studied in these experiments.

**Table 5.** Hyperparameter importance study for different hyperparameters across different datasets on SmartAugment experiments. For instance, for the RaVeNNa dataset, the probability of applying a data augmentation strategy is responsible for 46% of mean IoU's variability across the configuration space. The higher the importance value, the more potential it has to improve the performance for a given dataset.

| Dataset | p(aug) | col_mag | geo_mag | #col_ops | #geo_ops |
|---------|--------|---------|---------|----------|----------|
| KITTI | 0.13 | 0.12 | **0.24** | 0.14 | 0.03 |
| RaVeNNa | **0.46** | 0.04 | 0.06 | 0.06 | 0.05 |
| EM | **0.25** | 0.14 | 0.09 | 0.04 | 0.03 |
| ErFASst | 0.1 | 0.12 | 0.04 | **0.22** | 0.04 |

*4.4. Ablation Studies*

In addition to comparing our methods to the state-of-the-art approaches and the baseline, we report some ablation studies that give deeper insights into the impact of our methods.

### 4.4.1. RandAugment(++) Ablation Studies

To confirm that the improvement of SmartAugment over RandAugment++ comes from the method differences, we study RandAugment with different optimization methods. For this purpose, we compare classical RandAugment with Grid Search, RandAugment++ with Random Search, and RandAugment with Bayesian Optimization as optimization algorithms. We chose a cheap-to-evaluate dataset (EM) for this ablation study. As the results in Table 6 confirm, SmartAugment outperforms RandAugment, independent of the selected hyperparameter optimization algorithm. An interesting observation from the study is that RandAugment++ improves over the classical RandAugment as shown in Table 6. It is worthy to note that these improvement gains were achieved with fewer iterations and less computational costs.

Furthermore, Figure 3 shows that it can be sub-optimal to limit the number of applied augmentations to three, as is done in classical RandAugment. Therefore, increasing the search space as in RandAugment++ allows finding a better number of augmentation operations.

**Table 6.** Comparison of RandAugment variants with SmartAugment on the EM dataset. For each of the results, we took the test mean IoU of the best three performing configurations evaluated on the validation set. The results show that SmartAugment outperforms RandAugment(++), independent of the hyperparameter optimization algorithm.

| Method | HPO Algorithm | EM Dataset | # Iterations |
|--------|---------------|------------|--------------|
| Rand | Grid Search (classic approach) | 78.54 | 93 |
| Rand++ | Random Search | 78.83 | 50 |
| Rand++ | Bayesian Optimization | 78.84 | 50 |
| Smart | Bayesian Optimization | **79.04** | 50 |

4.4.2. SmartSamplingAugment Ablation Studies

In these ablation studies, we analyze the impact of annealing the probability hyperparameter over epochs and weighting the augmentation operations. For the experiments without annealing, we set the probability of augmentation $P$ to 1.

The results in Table 7 show that for three out of four datasets, annealing as well as weighting the augmentations are good design choices. Additionally, Table 7 shows that the combination of annealing the probability of augmentation and weighting the augmentations for RaVeNNa and ErFASst datasets improves the performance. Overall, SmartSamplingAugment does comparatively well and outperforms DefaultAugment and TrivialAugment across all datasets (see Table 1).

In the following, we give some possible explanations as to why annealing the augmentations for the EM dataset and weighting the augmentations for the KITTI dataset might not perform well. According to the hyperparameter importance study in Table 5, which quantifies the effect of how the values of these parameters affects the overall performance, the probability of augmenting data is the most important one for the EM dataset. Figure 7b, indicates that the EM dataset benefits from a high percentage of data augmentation; and therefore setting the probability of augmentation to 1 over the whole training yields better results rather than slowly increasing the probability of applied augmentations over the total number of epochs can be suboptimal. In contrast to the RaVeNNa dataset, where the probability of augmenting data is also an important hyperparameter (see Table 5), always augmenting data ($P = 1$) hurts performance. Figure 7a shows that annealing or progressively increasing the probability of augmentation for this particular dataset does seem to be a better alternative since we do early stopping.

**Table 7.** SmartSamplingAugment ablation study analyzing the impact of weighting the augmentations and annealing the probability hyperparameter over the whole epochs for different datasets. We evaluated each experiment three times using different seeds to obtain the mean IoU. For the experiments without annealing, we set the probability $P$ of applying augmentations to 1.

| Dataset | Weighting | | Without Weighting | |
|---|---|---|---|---|
| | Annealing | No-Annealing | Annealing | No-Annealing |
| KITTI | 66.53 | 67.15 | **67.49** | 67.13 |
| RaVeNNa | **90.72** | 87.07 | 85.65 | 85.68 |
| EM | 78.52 | **79.26** | 77.94 | 78.47 |
| ErFASst | **70.24** | 68.27 | 64.99 | 64.51 |

Furthermore, for the KITTI dataset, the geometric magnitude is the most important hyperparameter and Figure 9 shows that sampling a high geometric magnitude can hurt performance. In SmartSamplingAugment, rotation is strongly weighted, and there is a considerable probability that a higher magnitude for this operation is sampled. Figure 10 visualizes three KITTI and EM images, each rotated with a different magnitude, and gives an intuition why augmenting the KITTI dataset with high geometric operations can have a detrimental effect on performance. We note that we select the weights based on a study performed on a classification dataset, which probably is sub-optimal for semantic segmentation. However, this gives insight that studies focusing on optimizing the weights for augmentation operations can be a next step for further research.

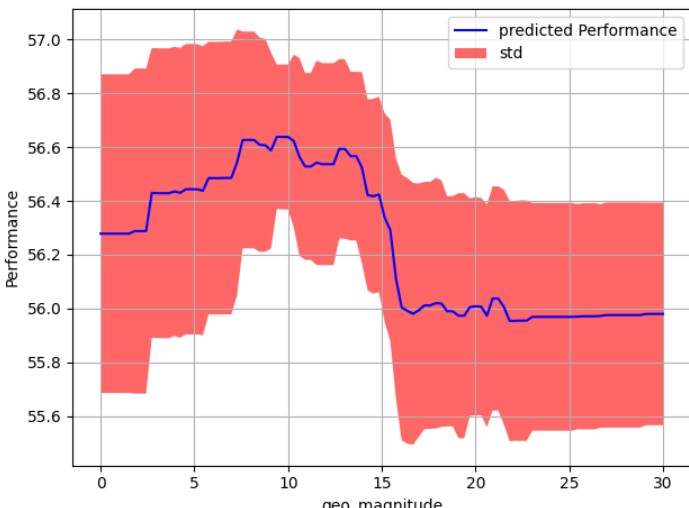

**Figure 9.** Results from SmartAugment on the KITTI dataset indicate that the geometric magnitude, which is the most important hyperparameter for this particular dataset, should be low. Taking this into account gives a possible explanation for why weighting the data augmentation with weights that focus on geometric augmentations might hurt performance (mean IoU) for the KITTI dataset.

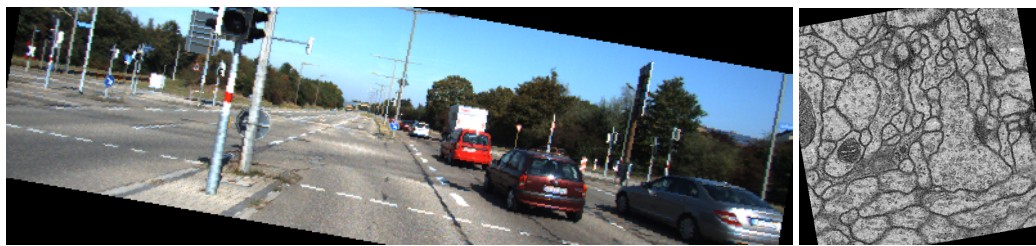

Rotation with magnitude 10

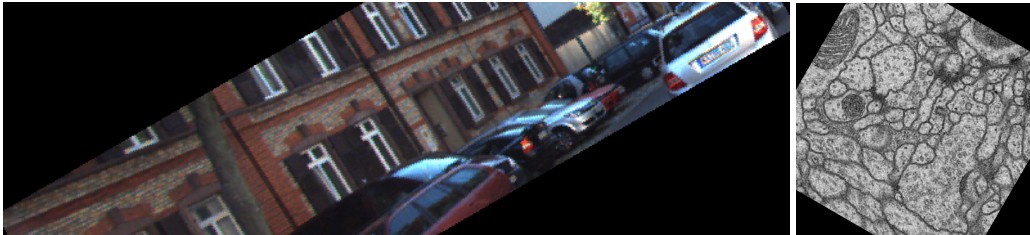

Rotation with magnitude 30

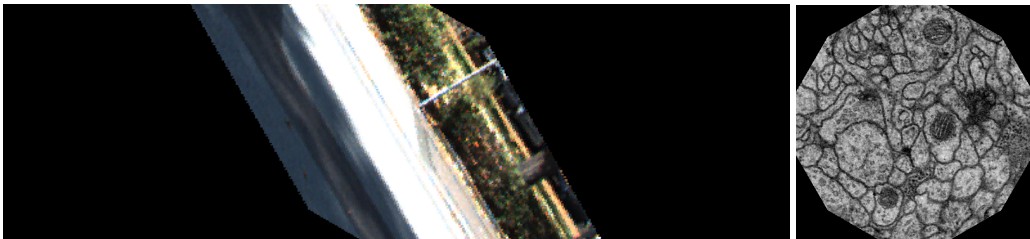

2× Rotation with magnitude 30

**Figure 10.** Visualization of three images from the KITTI (left) and EM (right) datasets, each rotated with a different magnitude.

## 5. Conclusions

In this work, we provide a first and extensive study of data augmentation for segmentation and introduce two novel approaches: SmartAugment, a new state-of-the-art method that finds the best configuration for data augmentation with hyperparameter optimization, and SmartSamplingAugment, a parameter-free, resource-efficient approach that performs competitively with the previous state-of-the-art approaches. Both methods achieve excellent results on different and diverse datasets.

With SmartAugment, we show that Bayesian Optimization can effectively find an optimal augmentation strategy from a search space where the number of color and geometric augmentations and their magnitudes are optimized separately, along with a probability hyperparameter for applying data augmentations. Our results show that the type of applied augmentation is essential in making good decisions for improved performance. Furthermore, a hyperparameter importance study indicates that the probability of applying a data augmentation strategy could have considerable responsibility for the mean IoU's variability across the configuration space.

With SmartSamplingAugment, we develop a simple and cheap-to-evaluate algorithm that weighs the augmentations and anneals augmentations to increase the percentage of augmented images systematically. The results show that this is a powerful and efficient approach that is competitive to the more resource-intensive approaches and outperforms TrivialAugment, a comparably cheap-to-evaluate method. Furthermore, SmartSamplingAugment opens the gate for more research on weighting and annealing data augmentation. A possible future work will study an extension of our methods to image classification, detection and 3D segmentation, particularly for biomedical applications.

**Author Contributions:** Conceptualization, M.N., D.W.; methodology, M.N., D.W.; software, M.N., D.W.; validation, M.N., D.W. and A.R.; investigation, M.N., D.W.; resources, A.R.; data curation, M.N.; writing—original draft preparation, M.N., D.W.; writing—review and editing, M.N., D.W. and A.R.; visualization, M.N., D.W.; supervision, A.R.; project administration, M.N., A.R.; funding acquisition, A.R. All authors have read and agreed to the published version of the manuscript.

**Funding:** This research was funded by the German Federal Ministry of Education and Research (13GW0203A) and approved by the local Ethical Committee of the University of Freiburg, Germany.

**Institutional Review Board Statement:** Not applicable.

**Informed Consent Statement:** Not applicable.

**Data Availability Statement:** The data used are referenced in the article.

**Acknowledgments:** The authors would like to thank the Department of Urology of the Faculty of Medicine in University of Freiburg for annotation of cystoscopic images that were used to build the RaVeNNA (Ravenna 4pi) dataset. Furthermore, thanks to Dominik Merkle for providing us with the ErfAsst dataset.

**Conflicts of Interest:** The authors declare no conflict of interest.

## Abbreviations

The following abbreviations are used in this manuscript:

| | |
|---|---|
| BO | Bayesian Optimization |
| IoU | Intersection over Union |
| SGD | Stochastic Gradient Descent |
| $N_C$ | Number of color augmentations |
| $N_G$ | Number of geometric augmentations |
| $M_C$ | Color magnitude |
| $M_G$ | Geometric magnitude |
| $P$ | Probability of applying augmentations |

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
