# Peer review of "Smart(Sampling)Augment: Optimal and Efficient Data Augmentation for Semantic Segmentation"

_algorithms, doi:10.3390/a15050165_

Round 1
Reviewer 1 Report
Smart(Sampling)Augment is proposed for data augmentation in segmentation tasks. For SmartAugment, hyperparameter optimization was performed. As described by authors, the proposed methods were validated in the different datasets of segmentation tasks.
Major points
“Algorithm 1” Please clarify the way to sampling from C_List and G_List. Random sampling?
Only number of iteration is described for computational cost in this paper. Please evaluate computational cost more accurately for Default, Rand++, Trivial, Smart, and SmartSampling. For example, time for one epoch training and time for one iteration.
Usefulness of the proposed method is not validated in classification and detection tasks. Please clarify this point as limitation.
Minor points
“# Iterations” in Table 1. Please clarify that “# Iterations” means number of iterations for hyperparameter optimization.
“more complex approaches such as CutMix [4].” Please cite Cutout and Mixup, here.
“Data augmentation has been applied to various areas, such as image classification [5], object detection [6], and semi-supervised learning [7].” Please cite the paper related to both data augmentation and segmentation. For example, please cite the following paper. https://www.mdpi.com/2076-3417/10/10/3360
“2. Related Work” Although the data augmentation in previous studies is described, hyperparameter optimization in previous studies is not described. Please cite and discuss previous papers related to both hyperparameter optimization with Bayesian Optimization and segmentation. For example, please cite the following paper. https://onlinelibrary.wiley.com/doi/10.1002/ima.22528
“apply C with M_C and G with M_G on I” Maybe, not “on” but “to”
“apply A_W with M on I” Maybe, not “on” but “to”
“Here we note that in SmartAugment, each augmentation can be sampled only once per image,” It seems that the combination of data augmentation operations is changed for each image in SmartAugment. If so, please clarify it.
“This default augmentation strategy is commonly used in semantic segmentation literature” In my opinion, it is difficult to determine default augmentation strategy. It may vary across the tasks.
Please describe the details of 4 datasets: number of images, number of classes, and class imbalance.
“split our datasets into training, validation, and test set.” Please clarify the number of images in training, validation, and test set of each dataset.
For many Figures, please clarify that mean IoU is used as the performance metric.
Reviewer 2 Report
The authors propose a first and extensive study of data augmentation for segmentation and introduce two novel approaches called SmartAugment (a method that finds the best configuration for data augmentation with hyperparameter optimization) and SmartSamplingAugment (a parameter-free, resource-efficient approach). Overall, a straight forward and interesting work. However, could you please discuss an extension to 3D, which would make the work very interesting for medical applications:
https://ieeexplore.ieee.org/document/8856297
Round 2
Reviewer 1 Report
Authors have addressed my concerns in the revision.